# Influence of Freeze-Thaw Aging on the Impact Performance of Damped Carbon Fiber Reinforced Plastics for Automotive Applications

Enrico Virgillito [1,*], Lorenzo Sisca [2] and Massimiliana Carello [2]

1   Applied Science and Technology Department, Politecnico di Torino, 10129 Torino, Italy
2   Mechanical and Aerospace Engineering Department, Politecnico di Torino, 10129 Torino, Italy;
    lorenzo.sisca@polito.it (L.S.); massimiliana.carello@polito.it (M.C.)
*   Correspondence: enrico.virgillito@polito.it

**Featured Application: Mass production of CFRP car door panels with NVH embedded system.**

**Abstract:** The increasing use of composite materials in the automotive field requires more attention with regards to the appearance of noise, vibration and harshness (NVH) study in cars construction. However, in car door panels production, impact characteristics need to be evaluated in sandwich laminates. Furthermore, it is important to consider the effect of prolonged environmental aging on crashworthiness properties. The innovative content of the work is the hygrothermal effects evaluation on impact performance for two damped CFRP sandwich laminates. In this paper, two damping materials, Kraibon HHZ9578/99 and SUT9609/24, were used as core between two skins of CFRP for sandwich composite production. Freeze-Thaw aging treatment according to IEC 60068, specific for Automotive, was performed to investigate environmental effects on components. Up to 750 h, it was demonstrated that water absorption is regulated by Fick's Law. The low-velocity impact behavior of the damped sandwiches has been studied according to ASTM D7136 throughout drop dart test equipment. Both main peak forces and energy absorption characteristics are negatively affected by aging condition. The introduction of damping core inside the composite structure of vehicle components can satisfy NVH constrictions. By contrast, at least same operating conditions must be assured in relation to not-damped components.

**Keywords:** NVH; CFRP; drop dart test; impact; ageing test; damping; rubber foil; automotive

## 1. Introduction

In the automotive industry, research is pushing for new and advanced solutions (especially for vehicle weight reduction). Weight reduction is a mandatory condition in mobility. Nowadays, it is mainly pursued thought two different methods: additive manufacturing (AM) and metal-to-PMC replacement. Weight reduction could be easily achieved by topological optimization of components [1]. Moreover, the creation of component with more than one function (for example cooling system and crash absorbers [2]) could be more and more explored in the really next future. In contrast, metal-to-PMC replacement is a reliable technology that is already mature enough to be introduced into the marketplace [3,4]. However, the mass introduction of PMC is largely taking place in car development. In this context, noise, vibration and harshness (NVH) characteristics improvement is pushing for the insertion of high damping material layers composites structures. In this sense, some attempt was already structurally developed and used [5]. Fasana et al. [6] have investigated the experimental NVH behavior of rubber damping layers when integrated in composite laminates, using the Oberst test. Subsequently, Fasana et al. [7,8] have applied the damped composites solution on a car door, evaluating the final performance compared to the whole composite. Stelldinger et al. [9] studied the impact damage resistance of

damped composites on tubes, evaluating that the positioning of the rubber in the outer region of the section leads to higher impact absorption energy. Roth et al. [10] proposed a new industrial solution to produce formed composite-metal hybrid laminates, utilizing the rubber layers to soften the deformation in the bended regions. Messana et al. [11,12] have applied the rubber layer in a multi-material Lower Control Arm (LCA) of a McPherson suspension, obtaining relevant advantages in terms of joining and damping effects from the component hybridization. By contrast, car door panel designer must ensure adequate crashworthiness for passive vehicle safety system.

A first approach to crashworthiness evaluation by drop dart test was studied since early 2000 for automotive passive safety system. A drop dart test was used to assess composite laminated structure energy absorption. For example, Belingardi et al. [13] analyzed the influence of laminates thickness and stacking sequence on energy absorption. The study of laminates thickness is strongly claimed in energy absorption. They defined two quantities for crashworthiness design approach are saturation energy (i.e., the minimum energy values for laminates penetration) and damage degree (i.e., the ratio between the total energy transformed (stored and dissipated) and the dissipated part of it). These two parameters can be useful for passive safety system design. In addition, no strain-rate effect was detected for GFRP. Similar results were discovered for CFRP composites [14]. Moreover, saturation energy value increases with the laminate thickness. The stacking sequence [0/90]i was detected as the best choice for energy absorption. In recent year, Boria et al. [15] investigated laminates thickness influence on energy absorption and fracture behavior for pure thermoplastic composite laminates (i.e., both the matrix and the reinforcement made of thermoplastic). A PP matrix reinforced with PP fiber was analyzed. Samples were composite laminated plates obtained overlaying [0–90]i plain weaves. The experimental results highlighted the influence of the thickness of the laminate on the impact behavior. With the considered low thickness, a perforation of the specimens was observed, whereas a ductile behavior and extended plasticity without a crack tip was obtained with the considered high thickness. The influence of the thickness on the deformation behavior was also confirmed examining the stiffness of the specimens during the impact. In contrast with previous literature, samples show high sensitivity to the impact energy level (considering the studied conditions). The impact damage was more extensive in the thin laminates than in the thick ones. Damage assessment is an important evaluation for the after-impact life of components. Malinowski et al. [16] evaluated damage by several techniques such as eye observation, back face relief, terahertz spectroscopy, laser vibrometry, X-ray microtomography, and microscopic observations. Except for microscopic observation, all of the other techniques can be categorized as non-destructive techniques (NDTs). However, just X-ray tomography can assure enough precision and accuracy for damage quantification. Virgillito et al. [17,18] confirmed this assumption. In fact, in their works, they compared energy absorption and damage quantification for GFRP by eye-observation, infrared observation, and X-ray tomography. Specific energy absorption (SEA) was calculated and compare among these NDTs.

However, the working environmental conditions must be taken into account as the fundamental boundary condition. In fact, unavoidable degradation processes could lead to catastrophic failure of the structure. A common scientific practice to evaluate the degradation state is the evaluation of moisture percentage retained by samples related to the hours of aging. Akay et al. [19] have correlated the water diffusion coefficient of the material and its degree of saturation to the degradation of the mechanical properties. Furthermore, Cysne Barbosa et al. [20] and Messana et al. [21] combined the mechanical tests respectively compression and tensile to the chemical analysis, such as FTIR spectroscopy, in order to evaluate the effects of hygrothermal and UV exposure on the polymer composites. In fact, it has been found that the degradation of the material at the fiber/matrix interface contributes to increase the fiber buckling and the matrix ductility.

Many literature articles report the impact characteristics of composite materials [13–18], but it is more difficult to find the evaluation of impact properties after aging. A first

approach was studied by Karasek et al. [22] in which the energy absorption property has been analyzed related to the effect of humidity and temperature. It was found that the variation of the composite behavior is mostly due to the modification of the Tg of the matrix. This characteristic is due to the strong plasticization that the matrix undergoes with liquids (not necessarily just water) [23]. Furthermore, the presence of dissolved salts and the temperature variation during the cycle tend to create micro-cracks due to the formation of solids: The matrix structure is affected by volume variation in the solid-liquid transition of liquid water to ice or by the salt's segregation from the liquid, due to a lowering of temperature and meniscus. Many researchers have focused on studying the response to the impact of aged composites in the climatic chamber. Its use has allowed to obtain more data on the impact performance of materials thanks to a greater speed of aging and the possibility of varying humidity and temperature at will [24–26]. Zhong et al. [27] stated that there is a change in the type of break, going from delamination to a set of debonding, fiber breakage and multiple cracking of the matrix, as the aging conditions change. Choi et al. [28] evaluated the parameters that influence aging (hygroscopic temperature, the ratio between matrix volume and volume of fiber, thickness, and lay-up) and so, the diffusion in the various directions. They arrived at the conclusion that the lay-up or the thickness do not influence the diffusion as much as the hygroscopic temperature and the presence of voids or the fiber/matrix interface. Gellert et al. [29] studied four different GFRPs aged in seawater in the laboratory, some loaded under set-strain for marine applications. A comparison between reinforced and non-reinforced materials were performed. Polyester, phenolic and vinylester was used as matrix while glass woven roving (630 g/m$^2$) as reinforcement. They demonstrated that flexural strength continued to degrade for the unloaded polyester and vinylester GRPs as water uptake continued toward saturation. By contrast, the unloaded phenolic lost 25% of initial strength at saturation, with no further loss as immersion continued from 200 to 800 days. Immersion ageing promote water up-taking respect to atmospheric one.

Experimental data of conventional materials for the industry are known from decades, but for innovative materials as composites it is not so obvious to find in literature data related to the aging. Some attempts on evaluation of impact on ageing PMC structures were performed both on pipes [30,31] and laminates flat specimens [32,33]. However, no attention was focused on NVH aspects. As consequence, it is not possible to find any information about ageing effect on high-damping layers PMC in scientific literature. Moreover, the presence of high-damping layers inside the composite could get the structure highly sensitive to aging due to the new interfaces. The main goal of the following study is to evaluate how hygrothermal aging influences the energy absorption during impact tests for two sandwich laminates obtained by a traditional composite (T300 carbon fiber with epoxy resin) integrating two types of high-strength damping materials (Kraiburg HHZ9578/99 and SUT9609/24). The phenomenon of diffusion was experimentally studied, measuring the relative moisture uptake as function of time. Second Fick's diffusion Law was applied to calculate the coefficient of diffusivity for all types of samples. Moreover, the evaluation of energy absorption and the electron microscopy analysis have been performed on the fiber-matrix interface and on the interfaces with the damping layers.

## 2. Materials and Methods

An IMP503 type Epoxy matrix reinforced with T300 Twill 2 × 2 carbon fibers (245 g/m$^2$) was used as reference material. Several specimens are fabricated as sandwich, including a core material made of a damping rubber, whose purpose is the vibration absorption improving of the composite. Two different damping materials were used, both supplied by Gummiwerk KRAIBURG GmbH: Kraibon HHZ9578/99 (HHZ) and Kraibon SUT9609/24 (SUT). The lamination of composite and damped sandwich were made by 8 layers at 0° of prepreg layup and damping rubber integration between the central layers. Plates 600 × 600 mm$^2$ were produced by vacuum bag method through autoclave for 3 h at 135 °C and 5 bar. The specimens were obtained from plates by water-jet cutting, avoiding the

thermal degradation of the polymer matrix. The specimens were cut in square plates $100 \times 100$ mm$^2$. Specimens thickness is related to the number of layer used. Then, T300/EP samples thickness was 2 mm while damping reinforce composites thickness (SUT and HHZ) was 2.8 mm.

In Table 1 are reported the averaged values of Young's modulus for the composite laminate, for the pure damping layers and for the Sandwich laminates with SUT and HHZ at temperature of −20, 20 and 60 °C in the untreated condition. The values have been measured using the Oberst test (ASTM E756). In this way, Young's modulus of specimens in axial direction could be obtained by a reliable alternative to tensile test. As can be noted, the Sandwich with SUT at room temperature demonstrates a lower Young's modulus than the Sandwich with HHZ.

**Table 1.** Young's Modulus averaged values for the composite laminate, for the pure damping layers and for the Sandwich laminates with SUT and HHZ at temperature of −20, 20, and 60 °C in untreated conditions.

| Temperature | Young's Modulus [GPa] | | | | |
|---|---|---|---|---|---|
| | **T300/EP** | **SUT** | **HHZ** | **Sandwich T300/EP + SUT** | **Sandwich T300/EP + HHZ** |
| −20 °C | 28 | 2 | 7 | 33 | 35 |
| +20 °C | 28 | 1 | 2 | 23 | 33 |
| +60 °C | 28 | 1 | 1 | 13 | 25 |

The impact tests were performed according the ASTM D7136 [27], which defines the standard for free-fall drop dart testing on composite materials. This test method is used for damage resistance evaluation. Three samples were tested for each condition. Independent variables studied in this paper were: rubber layer insertion (SUT or HHZ) on energy absorption; ageing effect on energy absorption; and dart velocity at impact on energy absorption. The damage resistance properties measured through this methodology is not an intrinsic property of the material, but it is influence by several factors. Dimensions and geometry of the specimen, sandwich layup, impactor geometry and settings, in addition to all boundary conditions, could strongly affect test results. The specimens are square plates $100 \times 100$ mm$^2$ constrained by clamping fixtures on a circular hole of diameter 76 mm. Tests were conducted in the range of low-velocity impact (LVI) using 6 m/s as drop dart velocity. In order to analyze the variation in energy absorption with velocity, 1.5 m/s were subsequently tested on sandwich damping structures. The same amount of initial kinetic energy has been imposed at 90 J changing the mass of the dart from 71 kg for 1.5 m/s to 6 kg for 6 m/s. The potential energy is considered without friction loss. The acquisition rate of 1 MHz is appropriate to collect an adequate number of points for a test that normally lasts less than 100 milliseconds. Vertical impact speed is then evaluated from the $\Delta t$ measured passing between two optical sensors. The calibration of the test velocity needs to be performed before starting with the activity specimens.

Dart velocity, displacement, and energy as a function of the time are calculated by the Equations (1)–(3):

$$v(t) = v_o + gt - \int_0^t \frac{F(t)}{w}\, dt \quad [\text{m/s}] \tag{1}$$

$$\delta(t) = \delta_o + v_o t + \frac{gt^2}{2} - \int_0^t \left( \int_0^t \frac{F(t)}{w}\, dt \right) dt \ [\text{m}] \tag{2}$$

$$E_a(t) = \frac{w(v_o{}^2 - v(t)^2)}{2} - wg\delta(t) \quad [\text{J}] \tag{3}$$

where: $v$ is dart velocity [m/s], $v_o$ is dart velocity at time = 0 s [m/s], $g$ is the is the free fall acceleration [m/s$^2$], $t$ is time [s], $F$ is load [N], $w$ is mass of the dart [Kg], $\delta$ is displacement [m], and $E_a$ is the absorbed energy [J].

The impact tests were carried out in first analysis on the untreated material and subsequently on the specimens aged for 750 h. No intermediate tests were performed due to the small number of specimens that can be inserted into the aging machine.

The aging has been carried out using an Angelantoni Challenge 250 Climatic Chamber. To simulate the behavior of the damping materials at different stages of life, it has been defined an aging cycle according to IEC 60068 standard [26]. The choice of this standard cycle is due to the specific application of these materials for Automotive sector. Output results are the average of at least three tests. The aging cycle used is reported in Figure 1, it has a duration of about 24 h, so each aging step of 250 h lasts for about 11 days. Specimens are exposed to different temperatures (−30 to +80 °C) at different stages of relative humidity (80 to 90% RH). For the different aging steps, the same specimens have been used. During freeze aging phase (blue dashed line), the relative humidity was not controlled as set by the IEC 60068 standard.

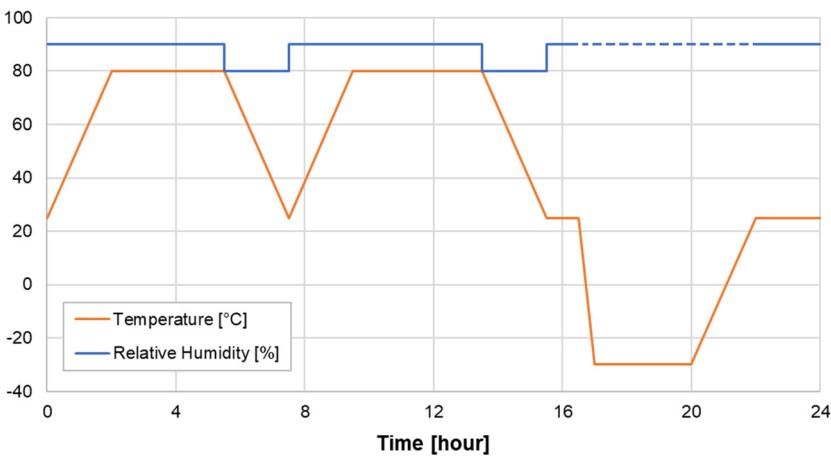

**Figure 1.** Controlled temperature and humidity diagram for the aging cycle IEC 60068.

The type of aging was chosen according to the final use of the component: an automotive industrial standard was used for this component. According to the IEC 60068 standard, aging method is composed by three cycles of 8 h each: two hot/wet cycles between 25–80 °C/90–80%RH and one cold cycle without humidity control. The alternation between cold and hot cycles is necessary to evaluate the variation of properties due to the formations of solid particles cooling the liquids. In fact, in each cycle there is a first phase of absorption of moisture during standstill at 80 °C and a subsequent ice formation at −30 °C. This alternation of environmental conditions could be modified the material in a technological condition that is no longer acceptable for its purpose. This is a factor of considerable importance for the design of energy absorption structures.

In the steady period at room temperature of the aging treatment, all of the specimens were dried and weighed using an analytic balance (with accuracy ± 0.0001 g), according to the ASTM D5229 [34], with the same methodology adopted in [35].

The increasing mass of water were normalized using Equation (4).

$$M_f = \frac{w_f - w_i}{w_i} \cdot 100\% \tag{4}$$

where: $M_f$ is the moisture mass uptake, $w_f$ is the final mass, and $w_i$ is the initial mass of the specimen.

The graph of relative moisture uptake in function of time was plotted for all of the configurations, damped and undamped. The coefficient of diffusivity, explained in the Fick's Second Law, was calculated in the initial linear zone using the Equation (5).

$$D_z = \pi \left( \frac{l}{4M_{eq}} \right)^2 \left( \frac{w_f - w_i}{\sqrt{t_f} - \sqrt{t_i}} \right)^2 \tag{5}$$

where: $D_z$ is the coefficient of diffusivity in z-direction (thickness), $M_{eq}$ is the moisture mass uptake at equilibrium, $l$ is the specimen thickness, $t_f$ is the final time, and $t_f$ is the initial time.

This is an important value to quantify the level of water saturation of the specimen. Greater is this number, greater is the absorption of water and the happening of aging effects on the interfaces and critical zones.

In order to investigate the adhesion of damping material to epoxy composite in the sandwich configuration, microscopic analysis was realized using a Field Emission Scanning Electron Microscope (Zeiss Supra TM 40), present in the laboratories of the Department of Applied Science and Technology (Politecnico di Torino). This technique allows to highly magnify a surface on a sample $5 \times 5 \text{ mm}^2$ and scan the morphology with an elevated depth of field. Two representative specimens of sandwich, not impacted and not aged, were cut along one side and polished on the surface of interest. Analyzed surfaces are sputter-coated with an Au layer before the examination in order to prevent electrons build-up charging.

## 3. Results

The experimental tests have been conducted on the carbon fiber specimens T300/EP with the configurations and quantity described in Table 2.

**Table 2.** Test configurations for T300/EP Composites and damped sandwiches.

| Test | Material | Damping Material | Test Specs | Condition before Test | N. of Specimens |
|------|----------|------------------|------------|-----------------------|-----------------|
| Absorption | T300/EP | Not Damped | IEC 60068 | untreated | 10 |
| | T300/EP | HHZ | IEC 60068 | untreated | 10 |
| | T300/EP | SUT | IEC 60068 | untreated | 10 |
| Impact | T300/EP | Not Damped | 6 m/s | untreated & aged | 6 |
| | T300/EP | HHZ | 1.5 & 6 m/s | untreated & aged | 12 |
| | T300/EP | SUT | 1.5 & 6 m/s | untreated & aged | 12 |

### 3.1. Aging Test and Absorption Model

All of the specimens, treated according to IEC 60068, were controlled in weight according to ASTM D5229 to establish the trend of moisture diffusion in the material. In Figure 2, the water uptake results according to IEC 60068 of the present paper are compared to water uptake results according to ASTM D5229.

The approach is the same, all of the data are collected at regular intervals for at least 400 h until the beginning of the asymptotic saturation. The difference is that ASTM D5229 provides a continuous spraying of distilled water on the specimen at 40 °C and 95%RH, whereas for the IEC 60068 the treatment is cyclic −20 to 80 °C and 80 to 90%RH.

The discrepancy of water uptake is appreciable between these two kinds of treatments, with the IEC 60068 the saturation is reached before.

Diffusion is a thermally activated process so, once a certain activation energy threshold is exceeded, the process depends on the temperature at which the system is located. This can be modeled by the variation of the diffusion coefficient $D$ according to the Arrhenius' law, in Equation (6):

$$D = D_0 \cdot exp^{\left(-\frac{\Delta E}{RT}\right)} \tag{6}$$

with $D_0$ as the maximal diffusion coefficient (at infinite temperature; m$^2$/s), $\Delta$E as activation energy for the diffusion process (J/mol), $R$ as the universal gas constant and $T$ as temperature (K). The ASTM D5229 thermal cycle keeps the temperature and humidity constant over time, therefore the total time of the duration of the diffusion process can be considered the sum of all of the hours of aging. In IEC 60068, on the other hand, the cycle is divided into three phases, two of which at 80 °C while the last at −20 °C. In this case it is correct to take into account only the hours in which the sample was subjected to 80 °C. It has been assumed that during freezing hours there is not enough energy to activate the diffusion process.

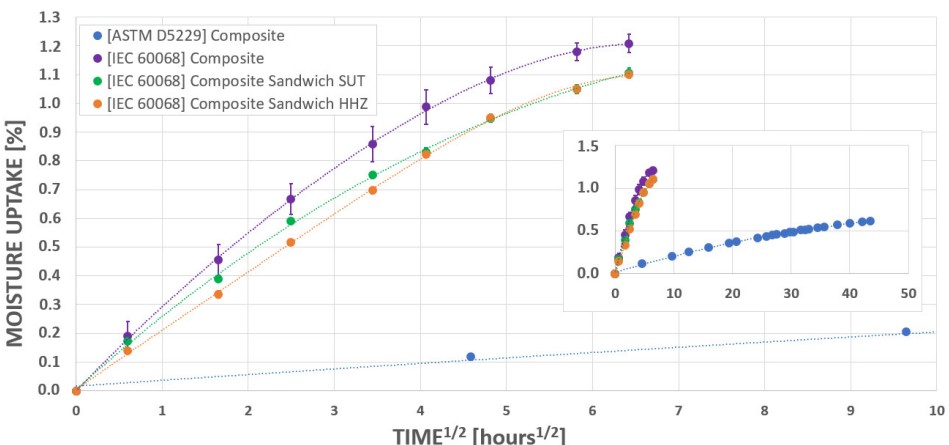

**Figure 2.** Moisture uptake for T300/EP composites and damped sandwiches with error bars.

Sandwiches with damping materials have the same behavior and the same moisture absorption, that is lower than the undamped composite. This can be due to the presence of the rubber, but the behavior has to be studied with further analysis. The diffusivity values for both the ASTM D5229 and IEC 60068 tests are reported in the Table 3. It can be seen that for the IEC 60068 not only the saturation is reached before the ASTM D5229, but also the coefficient of diffusivity is one order of magnitude greater.

**Table 3.** Moisture uptake and coefficient of diffusivity for T300/EP composites and damped sandwiches.

| Material | Standard | Moisture Uptake (%) | | Diffusivity Coefficient (m$^2$/s) |
|---|---|---|---|---|
| | | **M ($t$ = 25 h)** | **M∞** | **D** |
| T300/EP | Not Damped | ASTM D5229 | 0.103 | 0.7 | $1.9 \times 10^{-13}$ |
| T300/EP | Not Damped | IEC 60068 | 0.456 | 1.3 | $13.9 \times 10^{-13}$ |
| T300/EP | HHZ | IEC 60068 | 0.335 | 1.2 | $14.4 \times 10^{-13}$ |
| T300/EP | SUT | IEC 60068 | 0.390 | 1.3 | $12.2 \times 10^{-13}$ |

By comparing the data obtained with IEC 60068 and ASTM D5229 it is possible to obtain the activation energy of the diffusion process and the diffusion coefficient at infinite temperature for this kind of plastic-reinforced composite material. Plotting ln(D) vs. 1/T, it was obtained constant $D_0$ from the intercept and the value of $\Delta$E from the slope of the regression line, respectively. Thus, an activation energy equal to 237.3 J/mol and a maximal diffusion coefficient at infinite temperature of 6.2 m$^2$/s is obtained, which would allow one to obtain a diffusion profile at different temperatures with the consequent saturation absorption time.

### 3.2. Impact Test

In Figures 3–5 are plotted the energy-displacement mean curves for the untreated and aged specimens of the T300/EP Composite, the sandwich with Kraibon SUT and the sandwich with Kraibon HHZ. The error bars are useful to understand the reliability of the tests. The first graph compares the performance of the three configurations at 6 m/s. The following two graphs underline the velocity and aging effect on each damping material applied.

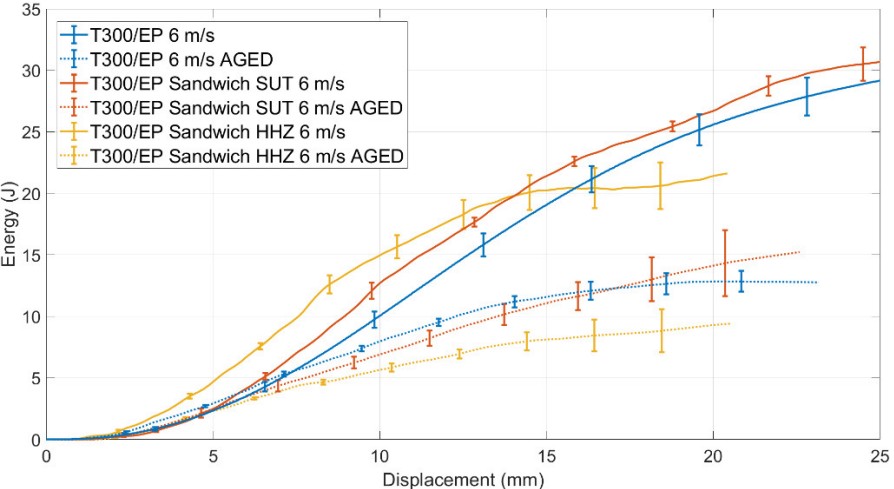

**Figure 3.** Energy-displacement mean curves for untreated and aged specimens of T300/EP composite.

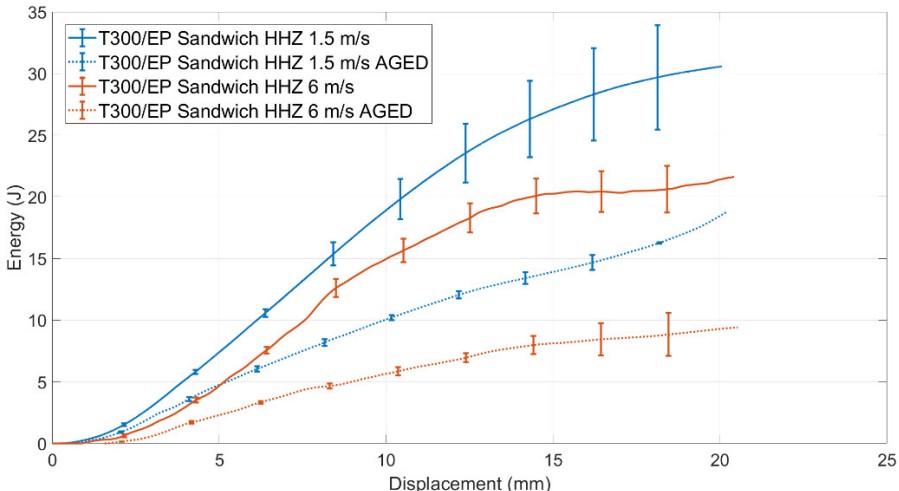

**Figure 4.** Energy-displacement mean curves for untreated or aged specimens of sandwich with Kraibon HHZ.

The averaged results with the standard deviation of the impact tests are reported in the Table 4.

In Figure 3, aged and not-aged impact performance of samples was illustrated. A distinct difference between the two types of reinforces can be noted. SUT maintains and slightly improves the performance of the reference material. On the other hand, HHZ layer reduces the energy absorbing properties of the composite structure. While in the first part of the impact test (until 15 mm of displacement), HHZ absorbed more energy respect to T300 and SUT samples, it failed at 20 mm of displacement prematurely by fragile rupture. Both these features can be explained by HHZ higher Young's modulus with respect to T300 and SUT.

**Table 4.** Averaged results with the standard deviation of the impact tests.

| Impact Test 90 J ASTM D7136 | | T300/EP | | Sandwich T300/EP + SUT | | | | Sandwich T300/EP + HHZ | | | |
|---|---|---|---|---|---|---|---|---|---|---|---|
| | | 6 m/s | | 1.5 m/s | | 6 m/s | | 1.5 m/s | | 6 m/s | |
| | | Untreated | Aged | Untreated | Aged | Untreated | Aged | Untreated | Aged | Untreated | Aged |
| Peak force [kN] | Avg. | 1.6 | 1.7 | 2.6 | 1.8 | 3.6 | 1.2 | 2.8 | 1.7 | 3.5 | 1.4 |
| | St.Dev. | 0.3 | 0.2 | 0.1 | 0.1 | 0.6 | 0.1 | 0.1 | 0.1 | 0.7 | 0.3 |
| Energy [J] | Avg. | 29.1 | 12.7 | 33.7 | 22.8 | 32.3 | 15.2 | 31.8 | 18.2 | 21.8 | 9.7 |
| | St.Dev. | 1.0 | 0.8 | 2.2 | 1.1 | 2.5 | 3.7 | 2.9 | 0.6 | 1.1 | 0.4 |

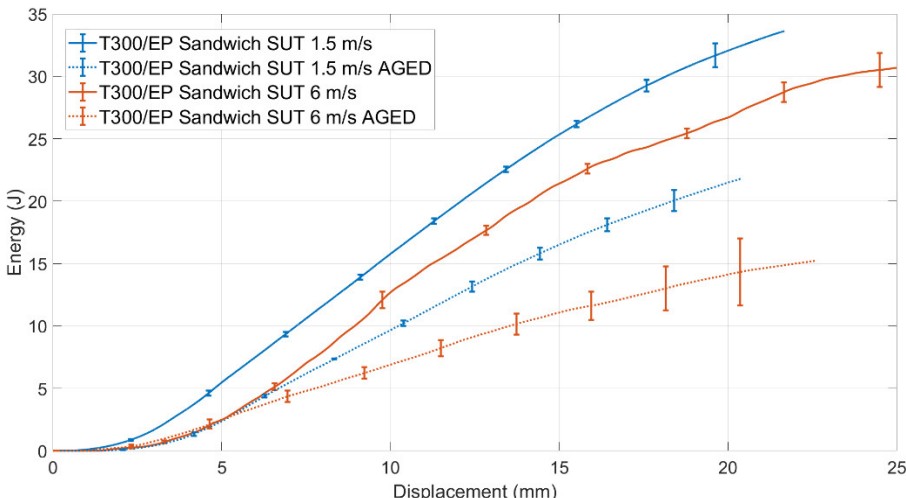

**Figure 5.** Energy-displacement mean curves for untreated or aged specimens of sandwich with Kraibon.

Similar tendencies are obtained as a result of aging. HHZ samples at 6 m/s, during an initial energy absorption phase, performed as the 1.5 m/s tested samples aged. There is therefore a similar effect between the increase in speed and the absorption of humidity. At 6 m/s aged instead there is an overlap of the effects (velocity and moisture) with a drastic lowering of absorbed energy. As can be seen in Figure 5, in the SUT samples, the overlapping of the effects is less marked in the initial phase of the test, since the 6 m/s AGED curve overlaps with the untreated and 1.5 m/s AGED curves. However, the comparison between the effect of speed and aging is exclusive to the first phase of energy absorption and does not continue for whole test. The variation in the values of peak force and energy are summarized in Table 5. The following considerations can be proposed:

1. Damping layers effect: The absolute values of peak force and energy at 6 m/s for untreated sandwich laminates were divided per the values of untreated composites. This value permits to consider the increment due to the integration of damping layer in the sandwich structure for the untreated condition;

2. Aging effect: The values of peak force and energy at 6 m/s for the aged materials were divided per the values of untreated respective configuration. This value permits to consider the effect of the aging for each material respect to the untreated condition;

3. Velocity effect: The ratio between the energy at 6 m/s and the energy at 1.5 m/s was found for aged and untreated specimens. This value permits to consider the variation of energy absorption at the two velocities for the untreated and aged conditions. It was evaluated that this calculation is not so effective for the peak force, due to the poor reliability of the maximum peaks in the load-displacement graphs.

**Table 5.** Relative variation of the impact performance, considering the effects of: damping, aging, and velocity.

| Impact Test—90 J ASTM D7136 | | T300/EP | Sandwich T300/EP + SUT | Sandwich T300/EP + HHZ |
|---|---|---|---|---|
| Damping Layers effect | $\Delta$Peak force (6 m/s) [%] | - | +125 | +118.8 |
| | $\Delta$Energy (6 m/s) [%] | - | +10.9 | $-25.1$ |
| Aging effect | $\Delta$Peak force (6 m/s) [%] | +6.3 | $-66.7$ | $-60.0$ |
| | $\Delta$Energy (6 m/s) [%] | $-56.3$ | $-52.9$ | $-55.5$ |
| Velocity effect | $E_6/E_{1.5}$ (Untreated) [%] | - | $-9.6$ | $-31.4$ |
| | $E_6/E_{1.5}$ (Aged) [%] | - | $-33.2$ | $-46.7$ |

In the velocity effect evaluation, it is possible to note that all of the untreated specimens show a reduction of absorbed energy. This fact could be attributed to strain-rate effects, as described in the literature [15,17]. However, it is interesting to note that SUT samples are more influenced by aging and speed than HHZ, despite the opposite seems during low energy absorption. In fact, in SUT samples, the loss of energy absorption capacity is tripled by the synergistic effect of aging and speed, while in HHZ samples it is only increased by 50%. The aging effect is uniform for all materials; a 50% decrease in energy absorbed at 6 m/s is noted. The addition of the elastomer layer involves an increase in Maximum Load (probably due to the increase in thickness of the composite) but there is an opposite effect in energy absorption between SUT and HHZ; the former increase their ability to absorb energy by 11% while the latter reduce it by 25%. This can be traced back to the interfaces created by the reinforcers within the composite.

### 3.3. Optical Observation and SEM Analyses

In Figure 6, some fracture zones of the sandwich specimens are shown. There is an evident delamination zone in fracture case of HHZ samples due to a low interfacial adhesion between the layers of damping material and composite.

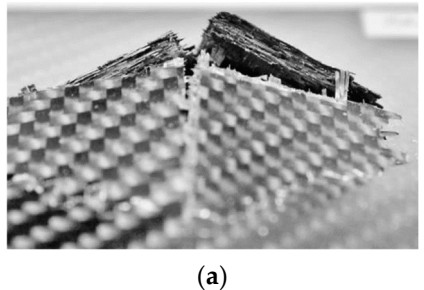 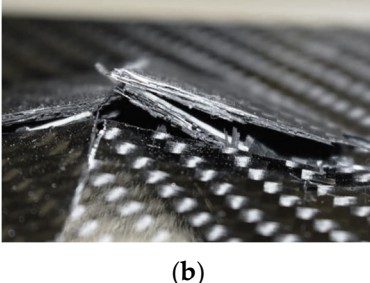

(**a**)    (**b**)

**Figure 6.** Breaking zones of sandwiches with damping material after the impact test: Kraibon SUT (**a**) and Kraibon HHZ (**b**).

In Figures 7 and 8 are reported the pictures of a cross section of two sandwiches damped with Kraibon SUT and Kraibon HHZ, both untreated and untested. For each image at 110× there is another magnitude in which the more interesting details are represented. The interface of Kraibon SUT appears continuous and well adherent with the faced layers, proving an excellent wettability. This feature could be one of the key reasons for the greater energy absorption of the composite damped with SUT. As a result of aging there are no substantial variations in the structure, which maintains excellent adhesion. In contrast, HHZ shows discontinuity and porosity within the damping layer. These defects have been found in the untreated sample and are therefore not attributed to treatment in temperature and humidity. These porosities are considered as preferential channels for water absorption.

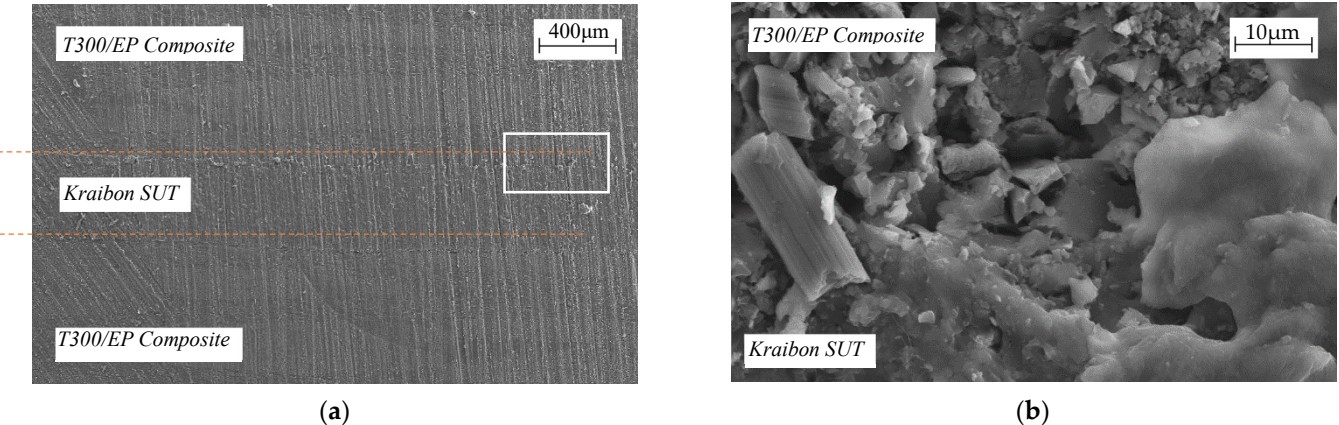

(a)                                                        (b)

**Figure 7.** SEM images of a sandwich damped with Kraibon SUT: magnitude (**a**) 110× and (**b**) 5000×.

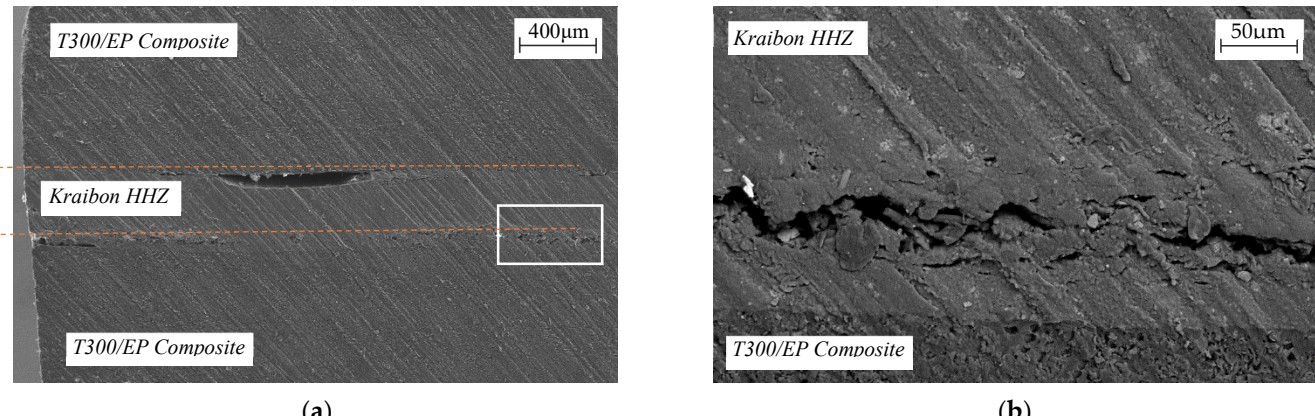

(a)                                                        (b)

**Figure 8.** SEM images of a sandwich damped with Kraibon HHZ: magnitude (**a**) 110× and (**b**) 1000×.

## 4. Discussion

The tests showed that the Kraibon SUT9609 damping material for each test has a greater ability to absorb energy compared to the Kraibon HHZ9578. This is due to the fact that the SUT damping layer is able to better adhere to the T300/EP layers in the production phase, guaranteeing a better stress distribution.

A strong tendency of Kraibon HHZ to delamination is evident from the specimen breaking type (Figure 6b). On the other side, the Kraibon SUT has a greater capacity for interlaminar adhesion since in none of the specimens is there a delamination on the interface, as confirmed by the SEM analyses (Figure 7a). Kraibon HHZ has a low capacity to adhere to carbon layers, which results in a worse energy absorption as shown in the Table 4. The energy in this case is dissipated during the delamination of the less cohesive layers, a phenomenon much less energetic than the breaking of the fibers and the matrix.

The greater interlaminar adhesion of the Kraibon SUT allows a greater ability to absorb energy. Regarding the Kraibon HHZ, the breakage of the fibersg appears fragmented, a characteristic that indicates a lower capacity to absorb energy. The Kraibon SUT also maintains the characteristic shape of the very similar curve during both impact velocities, which keeps the mechanical performance constant of the material at different speeds (Energy absorbed and peak of Max Load). In the case of the Kraibon HHZ the energy-displacement curve is considerably reduced translating into a weakening of the material with increasing speed.

The cause of the described behaviors of sandwich laminates is due to the mechanical characteristics of the pure damping materials. The Kraibon HHZ in fact, having a greater Young's modulus, tends to have a more fragile fracture, which leads delamination and

jagged rupture of the fibers. The Kraibon SUT having a lower Young's modulus allows to further support the fracture by absorbing a greater amount of energy.

## 5. Conclusions

In this paper, the mechanical characteristics of epoxy-based carbon composites with damping materials has been studied after prolonged environmental aging. Low velocity impact resistance was analyzed to assess the freeze-thaw aging treatment effects on multi-layered composites. Two different damping materials were used, Kraibon HHZ9578/99 and Kraibon SUT9609/24, and lamination processes and lay-up stack were maintained constant as T300/EP, taken as reference material. The impact tests were performed according the ASTM D7136 standard and effect of damping insertion, velocity, moisture and combination of them, are studied. SEM analysis were conducted to evaluate interfaces between composite and damping layer. The following conclusions can be drawn:

- It is possible to compare ASTM D5229 and IEC 60068 thermal cycle since diffusion is a thermal activated process. Diffusion during freeze phase could be taken as negligible; so just phase with enough thermal energy could be considered for time process evaluation.
- Activation energy of diffusional process was calculated and a maximal diffusion coefficient at infinite temperature is obtained; diffusion profile at different temperatures is now possible to plot in order to evaluate the saturation absorption time for these composites. Quite similar coefficient of diffusivity for all types of samples ensure the adoption of damping materials as functional layers do not influence the characteristics of sandwich structures after aging.
- The interface of Kraibon SUT appears continuous and well adherent with the faced layers, proving an excellent wettability. This is probably related to the lower modulus of elasticity and greater elongation at break than HHZ. This feature could be one of the key reasons for the greater energy absorption of the composite damped with SUT. As a result of aging there are no substantial variations in the structure, which maintains excellent adhesion. In contrast, HHZ shows discontinuity and porosity within the damping layer. These defects have been found in the untreated sample and are therefore not attributed to treatment in temperature and humidity. Certainly, bad interfacial adhesion and porosities are considered as preferential channels for diffusion.

As a result of the previous work, questions remain open on the chemical variations of the interface after aging or on the effect of the variation of stacking sequence, lay-up on the absorption of humidity and how this can influence the impact resistance. Further studies will be needed in the future.

**Author Contributions:** Conceptualization, E.V., L.S. and M.C.; methodology, E.V., L.S. and M.C.; validation, E.V. and L.S.; formal analysis, E.V.; investigation, E.V.; resources, M.C.; data curation, L.S.; writing—original draft preparation, E.V.; writing—review and editing, E.V., L.S. and M.C.; visualization, M.C.; supervision, M.C.; project administration, M.C.; funding acquisition, M.C. All authors have read and agreed to the published version of the manuscript.

**Funding:** This research received no external funding.

**Institutional Review Board Statement:** Not applicable.

**Informed Consent Statement:** Not applicable.

**Data Availability Statement:** The data presented in this study are available on request from the corresponding author.

**Acknowledgments:** The authors wish to acknowledge: Gummiwerk KRAIBURG GmbH for the supply of damping materials "KRAIBON® SUT9609/24-HHZ 9578/99", G.Angeloni® S.r.l. for the supply of CFRP material and SFC COMPOSITI S.r.l. for the production of damped composite panels and Ing. Alessandro Ferraris for the collaboration during the project.

**Conflicts of Interest:** The authors declare no conflict of interest.

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
