# Peer review of "Influence of Freeze-Thaw Aging on the Impact Performance of Damped Carbon Fiber Reinforced Plastics for Automotive Applications"

_applsci, doi:10.3390/app12084020_

Round 1

Reviewer 1 Report

Please provide mean square error in Figure 2.

Author Response

The Authors thank the reviewer for the suggestion. In the attachment, it will be found a complete point-by-point response to the reviewer’s comments.

Reviewer 2 Report

The paper discusses the influence of freeze-thaw aging on the impact performance of damped carbon fiber reinforced plastics for automotive applications.
The paper is required to consider the following comments before it can be considered for publication

1. The abstract needs to be rewritten to be more refletive. The write up should provide the aims/objective, methodology, key finding and practical implications

2. Introduction section needs to have flow and continuity between paragrapghs. The introduction section should lead the reader from a general perspective to be able to understand the research gaps that is being filled by the paper's objective

3. Paper requires moderate proof reading 

4. In section 3, methodology please provide a table to show the design of experiment (number of samples, independent variables, etc.)

5. In Figure 1 why is the aging cycle for temperature different after 14 hours?

Author Response

The Authors thank the reviewer for all the suggestions. In the attachment, it will be found a complete point-by-point response to the reviewer’s comments.

Reviewer 3 Report

Moderate English changes are required. For example check lines 41, 43, 160. Commas should be replaced with decimal in all tables and inside the manuscript. 

Line 146. About specimens, define testing and standard. Furthermore, the number of specimens tested. 

Line 194-196. How the authors made these calculations. Please elaborate.

Line 270. This assertions needs literature support, or further analysis

Author Response

(The authors gave the same response as above.)

Round 2

Reviewer 1 Report

Suggest to receive

Reviewer 2 Report

All comments were addressed.